# Key Parameters in the Manufacture of SiC-Based Composite Materials by Reactive Melt Infiltration

**DOI:** 10.3390/ma12152425

**Published:** 2019-07-30

**Authors:** Mario Caccia, Javier Narciso

**Affiliations:** 1Institute of Materials (IUMA), Alicante University, 03690 Alicante, Spain; 2Materials Engineering Department, Purdue University, West Lafayette, IN 47907, USA; 3Department of Inorganic Chemistry, Alicante University, 03690 Alicante, Spain; 4Center for Materials Science and Nanotechnology, Department of Chemistry, University of Oslo, 0316 Oslo, Norway

**Keywords:** SiC, CMCs, reactive infiltration

## Abstract

The manufacture of SiC-based composites is quite widespread, and currently different methods are employed to produce them. The most efficient method, taking into account the cost/performance ratio, is reactive melt infiltration. It consists in infiltrating liquid silicon into a porous preform that must contain carbon, so that SiC is produced during infiltration. In the present work, the synthesis of two SiC-based composite materials with very different applications and microstructures has been studied and optimized. In both cases, materials have been obtained with suitable properties for the selected applications. One of the materials studied is silicon carbide particles/silicon (SiC_p_/Si) for protection systems such as armor jackets, and the other one is carbon fiber/silicon carbide (C_f_/SiC) for use in braking systems. For the optimization, the dwell time and the atmosphere (Ar or primary vacuum) were used as variables. It has been found that in both preforms, the optimum conditions are 1 h dwell time and a vacuum atmosphere at 1450 °C. The effect of these parameters on microstructure and infiltration kinetics are discussed.

## 1. Introduction

Composite materials with a high volume fraction of SiC possess attractive properties for high-temperature applications (e.g., high stiffness, excellent corrosion and abrasion/erosion resistance, low thermal expansion coefficient and low density). This kind of composites, silicon carbide particles/silicon (SiC_p_/Si) in particular, have also shown an excellent behavior as armor systems, being able to absorb large amounts of energy upon an impact [1,2,3,4]. Their low density and their effectiveness as armor materials make them ideal candidates for ground transportation and personal armor devices. Nowadays, armor jackets are already being manufactured containing SiC_p_/Si composites. The most common arrangements of CMCs within bullet-proof jackets is in hexagons or squares. Furthermore, given the low coefficients of thermal expansion of SiC_p_/Si composites, they are excellent candidates for primary shields in future reusable spacecraft where they could protect against micrometeorites and other space particles’ impact, while also being able to withstand the atmosphere reentry process [5]. These kinds of composites can be easily manufactured with the reactive melt infiltration technique, which provides the possibility of synthesizing a great variety of geometries [6,7,8,9,10,11,12] due to its shape-preservation capabilities.

On the other hand, carbon fiber/silicon carbide (C_f_/SiC) composite materials, manufactured by reactive infiltration, have replaced conventional carbon steel brake discs in extreme applications such as airplanes and racing cars because of their superior performance, they lower density, and their higher wear resistance. This kind of composite materials has been tested by NASA under atmosphere re-entry conditions because of their potential use in space pods and future reusable space transportation systems (STS) [5]. They exhibit higher specific stiffness than some metal alloys and improved corrosion resistance compared to carbon/carbon composite materials. Furthermore, if coatings are applied, the corrosion by oxidation during re-entry to earth’s atmosphere can be substantially reduced.

The main objective of this work is to identify and optimize the most relevant experimental parameters, namely, dwell time and atmosphere, and to determine how these parameters affect the microstructure and the mechanical properties of such composite materials. For a better understanding of the infiltration process, an exhaustive characterization of the porous preform was made. Given their different chemical composition, the process limiting infiltration kinetics differs, thus affecting the optimization of the main infiltration parameters.

## 2. Materials and Methods

### 2.1. Preform Characterization

The porous preforms used in this work were supplied by different companies who are members of the 7th Framework European project High-Frequency Electromagnetic Technologies for Processing of Advanced materials and Graphite Expansion (H.E.L.M project) [13] and present diverse microstructures and pore systems. Their chemical composition was studied using X-ray diffraction (XRD, Bruker D8-Advance, Bruker) and thermogravimetric analysis (TGA, TGA/SDTA851e/LF/1600, Mettler Toledo), their density and porosity were determined by combining water and He picnometry (Acupyc 1330TC, Micrometrics), and their pore size distribution was obtained using Hg porosimetry (Poremaster-60, Quantachrome Instruments). The microstructure was studied using various microscopy techniques. A detailed description of the experimental techniques can be found in the previous works of our research groups [12,13,14].

#### 2.1.1. SiC_p_/C Preforms

SiC_p_/C preforms were kindly supplied by Petroceramics, and they are comprised of a bimodal mixture of α-SiC particles (280 and 1000 F.E.P. A grain size which is equivalent to 36.5 μm and 4.5 μm, respectively) bonded by a carbon matrix. These preforms were made by mechanically mixing α-SiC (black SiC, 99.5% purity) with 10 wt.% phenolic resin (Bakelite, Germany). The mixture was uniaxially pressed at 140 MPa and 150 °C to promote cross-linking of the resin. After that, the pressed mixture was pyrolyzed under a nitrogen atmosphere at 700 °C. Samples were supplied in the shape of bars with dimensions of 50 mm × 8 mm × 4 mm.

#### 2.1.2. C_f_/C Preforms

C_f_/C preforms were kindly supplied by BremboSGL group and are comprised of a mixture of chopped PAN-derived carbon fibers embedded in a carbon matrix. This preform was made by pressing a dry mixture of carbon fibers and phenolic resin (Bakelite, Germany) at 150 °C to promote cross-linking of the resin. Finally, pyrolysis of the product was performed at 900 °C to obtain a C_f_/C porous composite.

### 2.2. Reactive Infiltration and Infiltration Kinetics

Infiltration experiments were performed both under Ar flow and under vacuum. For such experiments, a Si bar was placed on top of the porous preform, which were then contained in a boron nitride (BN)-coated alumina boat-shaped crucible. The crucible was placed in the furnace and heated up to the target temperature of 1450 °C to melt the silicon. A detailed description of the procedure can be found in work previously published by our research group [14,15,16]. Because the experiments are performed using resistance furnaces, the heating rates that can be achieved (<8 °C/min) do not allow to accurately study the effect of temperature on infiltration. Since infiltration starts and proceeds rapidly after melting of Si (1410 °C), it is impossible to achieve isothermal conditions at higher temperatures without having some previous infiltration. For this reason, the experimental temperature was fixed to 1450 °C, slightly above the melting point of Si to ensure complete melting.

Because of the differences between SiC_p_/C and C_f_/C preforms, different behaviors during infiltration are expected. Since molten Si wets SiC, infiltration into the SiC_p_/C preform should not be limited by the chemical reaction between C and Si, and should proceed immediately after carbon dissolution. The laws of fluid dynamics will thus govern infiltration kinetics. Since the content in C in this preform is rather low, and is only intended to provide a good bonding between the reinforcement and the matrix, porosity and pore diameter should not change drastically during infiltration. On the other hand, in the C_f_/C preform, the content in reactive carbon is very high compared to the SiC_p_/C preform. Because molten Si does not wet carbon or carbon fibers, it is expected that in this preform, infiltration will be controlled by the chemical reaction at the infiltration front, and limited by the reaction kinetics. It should occur, therefore, much slower than for the SiC_p_/C preform. In this case, where reactivity is much higher, pore diameter reduction might play a major role in the infiltration kinetics.

### 2.3. Microstructure Characterization

The bulk density of the composites was measured using the Archimedes method. The solid density was determined using He picnometry (Acupyc 1330TC, Micrometrics). Residual porosity was calculated based on density measurements, as the relative difference between the solid and bulk densities. The microstructure of the obtained composites was studied using optical microscopy (PM3, Olympus), scanning electron microscopy (S3000N, Hitachi) and X-ray diffraction (D8-advanced, Bruker). Prior to microscopy characterization, cross-sections of the specimens were obtained, mounted in epoxy resin and polished to a 1 μm surface finish using standard metallographic procedures. For XRD, the specimens were ground to powder using a ball mill with tungsten carbide (WC) as milling media.

In the case of SiC_p_/Si composites, where the diffraction peaks of the β-SiC phase are coincidental with some diffraction peaks of the α-SiC phase, XRD does not allow for characterization of the reaction product. To overcome this issue, quantitative XRD was performed. In order to quantify the β-SiC formed during infiltration, a calibration curve was first obtained using α-SiC, β-SiC and Si powder mixtures with known compositions. The area ratio between the most intense common diffraction peak for both phases (diffraction peak for the 111 crystalline plane at 35.65°) and the second most intense peak of the α-SiC phase, which is not seen in the β-SiC phase (diffraction peak of the 103 crystalline plane at 38.2°) was calculated and plotted against the molar ratio between α-SiC and β-SiC (X_αSiC_/X_βSiC_) to obtain the calibration curve. This calibration curve was then used to estimate the X_αSiC_/X_βSiC_ in the specimens.

To obtain the phase quantification of C_f_/SiC composites, a selective etching of each phase was performed with successive weight measurements. To remove the carbon fibers and the amorphous carbon, the samples were heat treated at 800 °C in air during 9 h. To remove silicon, the remaining sample was treated with concentrated hydrofluoric acid (HF 40%) at 80 °C during 12 h. SiC was obtained as the difference between the original weight and what remained.

### 2.4. Mechanical Characterization

The mechanical properties most relevant to the potential applications were measured for each composite. For the SiC_p_/Si composites, bending and compressive strengths were measured in a Universal Testing Machine (model 4411, Instron) with the 3-point bend test and the quasi-static compression test configurations following the UNE-EN-843-1:2006 standard. The 3-point bending test was performed at a cross-head displacement of 0.1 mm/min (strain rate of 10^−3^ s^−1^) and the compression test was carried out at a strain rate of 10^−3^ s^−1^. An extensometer was used during the bend test, and the flexural modulus was calculated from these results. For C_f_/SiC composites, the bending strength was measured following the same procedure described for the SiC_p_/Si specimens. The wear behavior of this material was studied using the ball-on-disc setup (CSEM Instruments built according to ASTM G99 standard). The wear test was performed at a linear speed of 10 cm/s with a load of 5 N and a duration of 10 km. The average friction coefficient was measured from this test. The average Vickers microhardness was also measured using a microindentor (Micromet 2100, Buehler) according to the ASTM C1327 standard. Vickers microhardness was measured using a load of 1 kg with a holding time of 15 s.

## 3. Results and Discussion

### 3.1. Preform Characterization

Table 1 summarizes the results of the characterization of the SiC_p_/C preform.

The composition was obtained by TGA in an oxidizing atmosphere (N_2_:O_2_ = 4:1) in the range 25–800 °C and it is 95.3 wt.% SiC and 4.7 wt.% C. The corresponding thermogram is shown in Figure 1a where the first initial mass loss corresponds to moisture in the sample, and the mass loss at 600 °C corresponds to the combustion of the C phase. The remaining mass corresponds to SiC, which remains unaltered in this temperature range. XRD tests indicated that α-SiC is the only crystalline phase present in the preform (see Figure 1b), exhibiting a mixture of the 4H and 6H polytypes.

The bulk density of the preform ranges from 2.10–2.20 g/cm^3^, while the He density is 3.10 g/cm^3^. Considering these values, the open porosity was calculated, yielding a value of around 30%.

Figure 2a shows an optical microscope image of a polished cross-section of the preform, where the α-SiC phase is depicted in white, the C phase in grey and the porosity in black. A homogeneous distribution of the phases and the porosity is clearly observed.

Mercury porosimetry showed a narrow pore size distribution around a pore size of 1 micron (P = 1 MPa). The intrusion curve and the pore size distribution derived from it are shown in Figure 2b.

Table 2 summarizes the results of the characterization of the C_f_/C preform.

The composition was obtained by TGA in oxidizing atmosphere (N_2_:O_2_ = 4:1) in the range 25–900 °C and it is 70 wt.% carbon fiber and 30 wt.% carbon binder. The corresponding thermogram is shown in Figure 3a, where the first initial mass loss corresponds to moisture in the sample, and the mass loss at 600 °C corresponds to the combustion of the amorphous C binder (product of pyrolysis of the phenolic resin). The final mass loss observed at 750 °C corresponds to the combustion of the PAN-derived carbon fibers, which are thermally more stable than the amorphous C binder. XRD tests indicated that carbon fibers are the only crystalline phase (See Figure 3b). However, they present low crystallinity (very broad diffraction peak) as the spacing between graphitic layers calculated from the position of the 0002 peak (0.35 nm) is far from the theoretical one for perfect graphite (0.33 nm).

The bulk density of the preform ranges from 1.20–1.25 g/cm^3^, while the He density is 1.83 g/cm^3^. Considering these values, open porosity was calculated yielding a value of around 30%.

Figure 4a shows an optical microscope image of a polished cross-section of the preform, where the cross-section of the carbon fibers is easily distinguished as they appear in the shape of circles or ellipses, depending on the direction of the fiber. By looking at the microstructure, it becomes clear that fibers are randomly oriented and grouped in fiber bundles. The carbon binder is observed surrounding the fibers and fiber bundles, and porosity of different sizes is distinguished.

Figure 4b shows the intrusion curve and the pore size distribution obtained from mercury porosimetry. A broad distribution of pore sizes around two values, 2 and 10 μm, is observed. This is in agreement with the porosity observed in Figure 4a. The pores with sizes around 2 μm correspond to the interfiber spaces, while the pores with sizes around 10 μm correspond to the interbundle spaces. The high initial intrusion can be attributed to the filling between the pieces of sample rather than to actual porosity of the material.

### 3.2. Infiltration of SiC_p_/C and C_f_/C Porous Preforms

The time required for full infiltration of a typical 5 mm-thick preform was calculated for both preforms using Darcy’s Equation:$$h^{2}=\frac{2\cdot K\cdot t}{\eta \cdot \Phi }\cdot \Delta P$$
where *h* is the infiltrated distance, *K* is the permeability, *t* is the time, η is the fluid dynamic viscosity, Φ is the porosity of the system and Δ*P* the pressure drop that drives the infiltration. The permeability is proportional to the pore radius according to:
$$K=a\cdot r^{2}$$
where a is a proportionality constant and *r* is the average pore radius. In the case of systems where the liquid wets the solid and infiltration proceeds spontaneously (*θ* < 90°), the pressure drop is the capillary pressure (*P_c_*) given by:$$P_{c}=\frac{2\cdot \lambda \cdot \gamma_{LV}\cdot cos\theta \cdot \left(1-\Phi \right)}{r\cdot \Phi }$$
where *λ* is a geometrical factor, $\gamma_{LV}$ is the surface tension of the liquid, *θ* is the contact angle of the liquid on the solid, Φ is the porosity of the system, and *r* is the average pore radius. Table 3 shows the parameters for the SiC systems used to calculate the infiltration time.

This model suggests that vertical infiltration of a preform of 5 mm thickness should be achieved within a fraction of seconds, 0.54 s for the SiC_p_/C preform and 0.05 s for the C_f_/C preform. However, it must be considered that the time required for full infiltration of a 3D sample is greater, since the melt has to spread in the x- and z-directions, and usually, permeability in the x-direction is a couple of orders of magnitude lower for uniaxially pressed materials than in the z-direction [24]. Nevertheless, this model is likely to predict a time for full infiltration of a couple of orders of magnitude faster than the real one, since infiltration in the C_f_/C preform should be limited by the chemical reaction at the infiltration front. Furthermore, none of these models take into account the pore reduction or closure phenomena, which might play a major role in infiltration kinetics. The infiltration time calculated for the SiC_p_/C preform should be closer to the real infiltration times since the chemical reaction should not delay infiltration as much as in the C_f_/C preform.

Figure 5 shows the evolution of bulk density of the infiltrated specimens with different dwell times in the furnace at 1450 °C for both kinds of preforms under Ar flow (black dots) and under a vacuum atmosphere (white dots). The red and blue lines are guides to the eye.

In general, lower densities are obtained when infiltrating under Ar flow, most likely because of gas entrapment within the pores of the preform. Because of the set up used for infiltration, the surface of the preform can be rapidly covered with Si after it melts, preventing gas inside the pores from exiting the preform (see Figure 6).

The fact that longer dwell times yield denser materials provides more evidence of this phenomenon, as gas bubbles can diffuse through the melt and out of the preform if given sufficient time. Further proof of gas bubbles entrapment is provided by the evolution of bulk density with dwell time obtained for infiltrations performed under vacuum. Not only are higher values of bulk density are obtained but also a density value that is close to the theoretical maximum density is obtained after 1 h at 1450 °C. The huge difference in density between the sample infiltrated for 1 h in Ar and under vacuum can only be explained because of gas remaining within the microstructure for the sample infiltrated in Ar, as models suggest that full vertical infiltration should occur in the first few minutes.

The difference between densities achieved under Ar or vacuum is smaller for the C_f_/SiC composite than for SiC/C (see Figure 5). Because pores are wider for the C_f_/C preform, and spreading of liquid Si and thus infiltration proceeds slower because they are limited by the chemical reaction at the infiltration front, gas can escape the pores more easily as they are filled. Nevertheless, the slightly higher densities obtained with vacuum suggest that gas can remain trapped within the smallest pores of the preform during infiltration.

### 3.3. Microstructure of the Composites.

#### 3.3.1. SiC_p_/Si Composite Materials

Figure 7 shows a representative optical micrograph of a polished cross-section of a fully infiltrated SiC_p_/Si composite material at two different magnifications. The porous carbon matrix of the SiC_p_/C preform has been replaced with a dense Si matrix, and the bimodal SiC particle distribution is clearly observed.

Figure 8 shows the evolution of microstructure at different dwell times for samples infiltrated under Ar flow. Full vertical infiltration has been achieved for all samples. However, samples show large residual open porosity, particularly, the sample infiltrated with a dwell time of 1 h. The microstructures support the idea that the lower values of density obtained under Ar flow are a consequence of gas entrapment within the microstructure of the material. Consequently, even if full infiltration is observed, large pores remain within the composite. This porosity decreases with increasing dwell time as bubbles diffuse out of the preform.

Further evidence of this phenomenon is given in Figure 9, where the evolution of the microstructure of SiC_p_/Si composite materials infiltrated under vacuum is shown. No differences in the microstructure with dwell time are observed for samples infiltrated under vacuum. Furthermore, the absence of the large pores observed for samples infiltrated under Ar confirms that gas entrapment is responsible for the lower densities of those samples.

Quantitative XRD analysis yielded β-SiC/α-SiC molar ratios between 0.05–0.2, which is consistent with the initial amount of C having fully reacted. The rather high dispersion in the beta/alfa molar ratios measured can be mainly attributed to some heterogeneity in the C binder distribution within the preform, as well as to the dispersion in the A_111_/A_100_ of the standards used to prepare the calibration curve. The generated calibration curve for X-ray quantitative analysis is shown in Figure 10.

#### 3.3.2. C_f_/SiC Composite Materials

Figure 11 shows two representative micrographs obtained with optical microscopy of polished cross-sections of C_f_/SiC fully infiltrated specimens at different magnifications. A total of 4 different phases are observed within the microstructure, namely, unreacted amorphous carbon (dark brown phase), carbon fibers (dark brown, circle or ellipse cross-section), unreacted silicon (white), and β-SiC (light brown, faceted crystals). The residual porosity is depicted in black. The microstructure of the preform is mimicked after infiltration but amorphous carbon has been partially dissolved and is now covered in a continues layer of SiC. Silicon remains trapped within the residual porosity and carbon fibers remain unaltered, with some minor amounts of fibers having been dissolved.

As discussed in the previous section, because Si does not wet carbon or carbon fibers, infiltration in this preform should be limited by the chemical reaction at the infiltration front, and should, therefore, occur much slower than predicted by Darcy’s law. Furthermore, because of the large content in amorphous reactive carbon, pore reduction should play a major role in infiltration kinetics. Figure 12 shows evidence of severe pore reduction in C_f_/SiC samples where even pore closure is observed for smaller channels.

There were no large differences observed in the microstructure between samples infiltrated in Ar and under vacuum, besides some additional residual porosity. After the SiC layer has covered the amorphous carbon and the carbon fibers, its growth is controlled by diffusion of C and Si through the SiC interface, and the growth kinetics become much slower, therefore, no large differences are observed within the dwell times tested. There was some increase in carbon fiber dissolution with dwell time, although, as shown in Figure 13, this was particularly relevant for samples infiltrated with a dwell time of 5 and 8 h. This phenomenon could affect the mechanical properties of the material negatively.

Figure 14 shows the interface formed between the carbon fiber bundles and the SiC and amorphous carbon phases. An intimate contact between fibers and the amorphous carbon phase is observed. However, this interface is rather weak and does not provide enough mechanical resistance. The contact between the fibers and the SiC phase is defective and is achieved through an interface that comprises dissolution pockets formed during early stages of infiltration and the nano-SiC that precipitates upon cooling of the C-supersaturated Si within the dissolution pockets [14].

Table 4 shows the calculated composition after the selective etching. As expected, no differences in the amount of SiC phase are observed, yielding contents in SiC between 35–38%, independently of the dwell time. The small differences are attributed to the heterogeneous distribution of the phases in the preform.

### 3.4. Mechanical Properties of Composites Materials

After studying the effects of dwell time and atmosphere on infiltration and microstructure of the materials, the optimum conditions were selected for each preform based on the quality of the material and the shortest processing time. Mechanical properties of the produced composites under optimum infiltration conditions were tested, selecting the testing methods according to the potential applications of each composite material. The optimum infiltration conditions for the SiC_p_/C preforms were selected as 1450 °C, with a dwell time of 1 h under a vacuum atmosphere. The optimum infiltration conditions for C_f_/C preforms were selected as 1450 °C, with a dwell time of 1 h under a vacuum atmosphere.

#### 3.4.1. Mechanical Properties of SiC_p_/Si Composite Material

Table 5 summarizes the measured mechanical properties for the SiC_p_/Si composite material. The mechanical performance of the material was characterized using the 3-point bending test and the quasi-static compression test. As expected for a ceramic material, the SiC_p_/Si composite exhibits higher compressive resistance than flexural, where the fracture is dominated by matrix cracking on the tensile face. The material exhibits a high specific strength (bending strength divided by density) compared to some metals and metal alloys [25,26].

Figure 15 shows SEM images of the fracture surfaces of the material after mechanical testing, where it is possible to observe that the main fracture mechanism of the material subjected to flexural stresses is a mixture of interparticle/intraparticle mechanisms, as most SiC particles appear broken (indicated by the small black arrows). On the other hand, the main fracture mechanism of the material subjected to compressive stresses is an interparticle mechanism (matrix cracking).

#### 3.4.2. Mechanical Properties of The C_f_/SiC Composite Material

Table 6 summarizes the mechanical properties measured for the C_f_/SiC composite material. The mechanical performance of the material was characterized using the 3-point bending test, the Vickers microhardness test, and the ball-on-disc test.

The material exhibits a high surface hardness and a high friction coefficient, coupled with a low density. The rather low bending strength and elastic modulus can be explained when analyzing the microstructure, where carbon fibers are bonded to the SiC phase through a weak interface that comprises dissolution pockets and nano-SiC precipitated upon cooling in those C-supersaturated dissolution pockets. The mechanical properties exhibited by the material make it an excellent candidate for high-performance braking systems. The analysis of the fracture surface after the bending test indicated that the main fracture mechanism is matrix cracking and fiber pullout, and showed that fibers are pulled out in bundles rather than individually, explaining the low bending strength and elastic modulus values registered. Bundles are pulled through the weak C_f_/C interface and remain bonded within the bundle by a stronger C_f_/SiC interface, as shown in Figure 16.

## 4. Conclusions

Different porous carbon-containing preforms have been studied and characterized as candidates to be used in the synthesis of SiC-based composite materials by reactive infiltration. The infiltration process was successfully achieved and C_f_/SiC and SiC_p_/Si composite materials were obtained. The variables of atmosphere and dwell time were optimized for each preform, selecting a dwell time of 1 h and vacuum atmosphere as the optimum conditions for infiltration of both preforms at 1450 °C. The most relevant mechanical properties for the applications of the different composite materials were measured. The C_f_/SiC material shows outstanding properties to be used in ground and air transportation braking systems and could be used potentially in future reusable space transportation. The SiC_p_/Si material shows excellent properties for use in protection systems such as armored jackets or armored vehicles. Owing to its thermal properties, it could be implemented as shields in space vehicles to protect against minor impacts.

## Figures and Tables

**Figure 1 materials-12-02425-f001:**
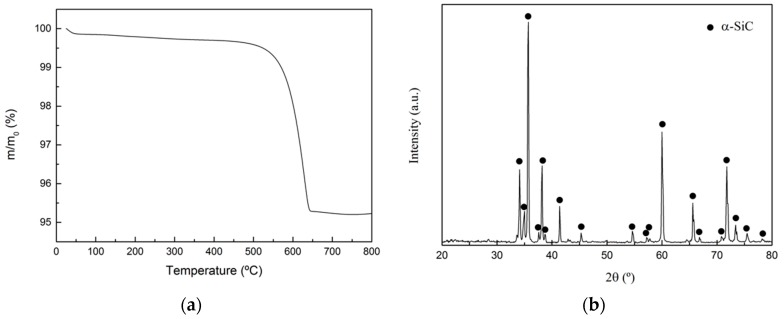
(**a**) Thermogram of the SiC_p_/C preform in an oxidizing atmosphere (N_2_:O_2_ = 4:1); (**b**) XRD pattern for the SiC_p_/C preform.

**Figure 2 materials-12-02425-f002:**
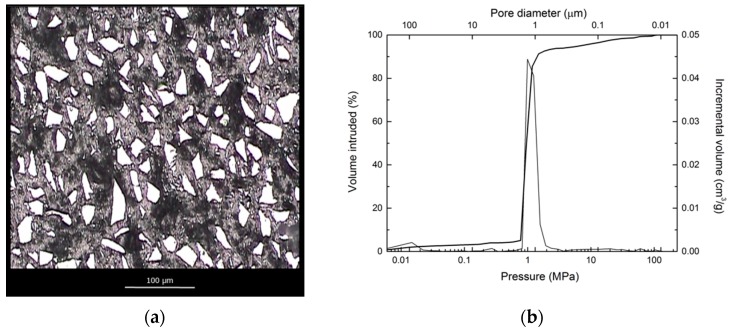
(**a**) Optical microscopy image of a polished cross-section of the SiC_p_/C preform; (**b**) mercury intrusion curve as a function of pressure and pore size distribution for the SiC_p_/C preform.

**Figure 3 materials-12-02425-f003:**
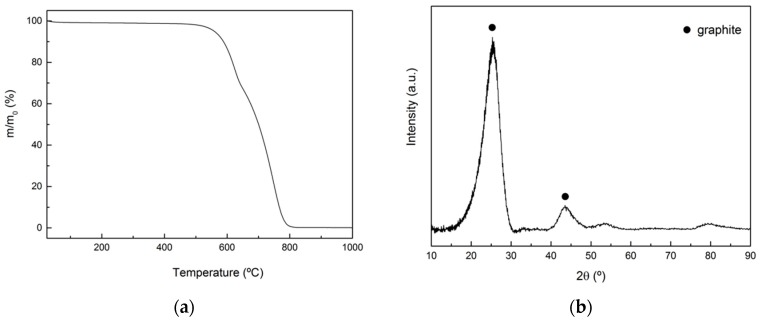
(**a**)Thermogram of the C_f_/C preform in oxidizing atmosphere (N_2_:O_2_ = 4:1); (**b**) XRD pattern for the C_f_/C preform.

**Figure 4 materials-12-02425-f004:**
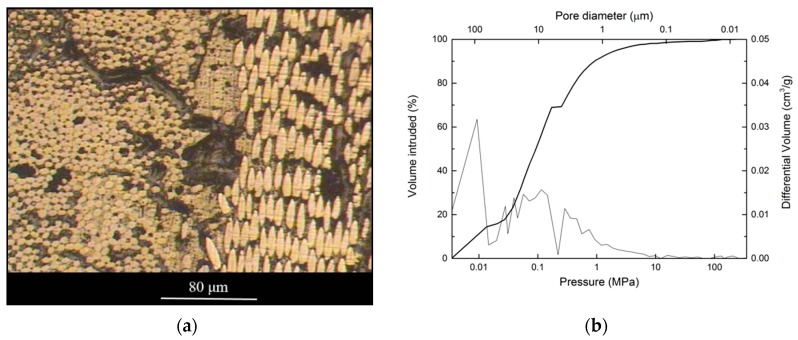
(**a**) Optical microscopy image of a polished cross-section of the C_f_/C preform; (**b**) mercury intrusion curve as a function of pressure and pore size distribution for the C_f_/C preform.

**Figure 5 materials-12-02425-f005:**
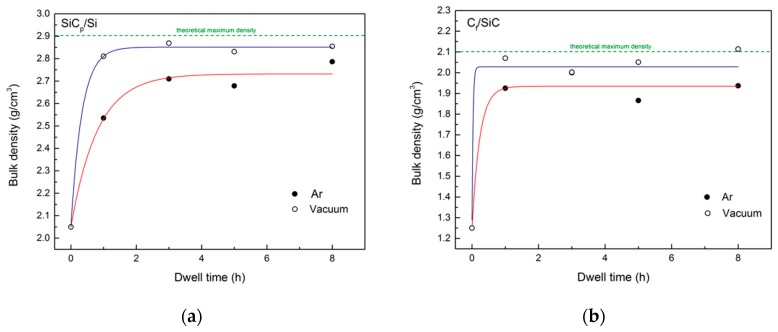
Bulk density as a function of dwell time at 1450 °C under Ar and vacuum atmospheres of SiC_p_/C (**a**) preforms and C_f_/C preforms (**b**).

**Figure 6 materials-12-02425-f006:**
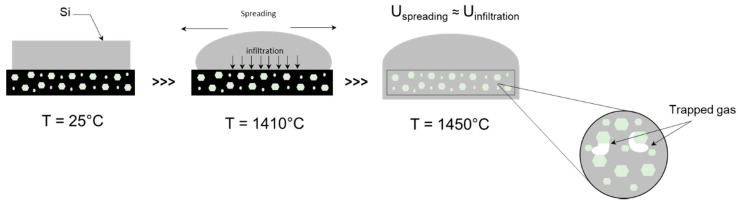
Schematic representation of the infiltration process in SiC_p_/C preforms under Ar flow.

**Figure 7 materials-12-02425-f007:**
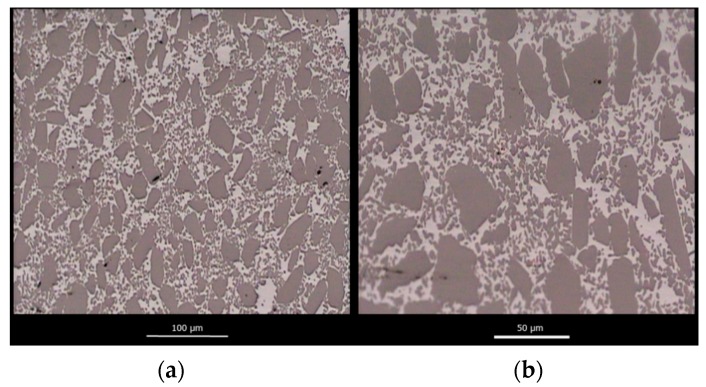
Representative optic micrographs of a fully infiltrated SiC_p_/Si composite material with a low (**a**) and a high (**b**) magnification.

**Figure 8 materials-12-02425-f008:**
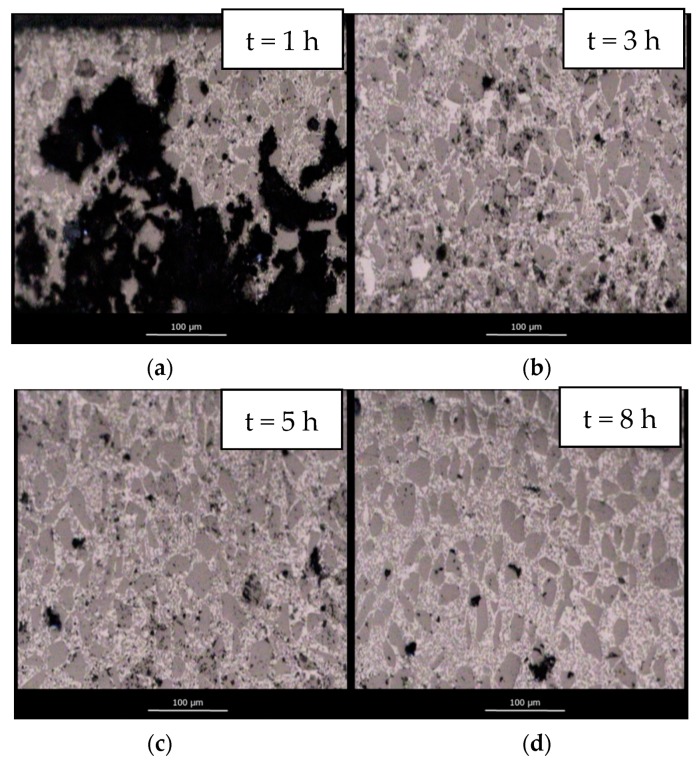
Evolution of the microstructure of SiC_p_/Si composite materials infiltrated at 1450 °C in an Ar flow atmosphere with a 1 h (**a**); 3 h (**b**); 5 h (**c**) and 8 h (**d**) dwell time.

**Figure 9 materials-12-02425-f009:**
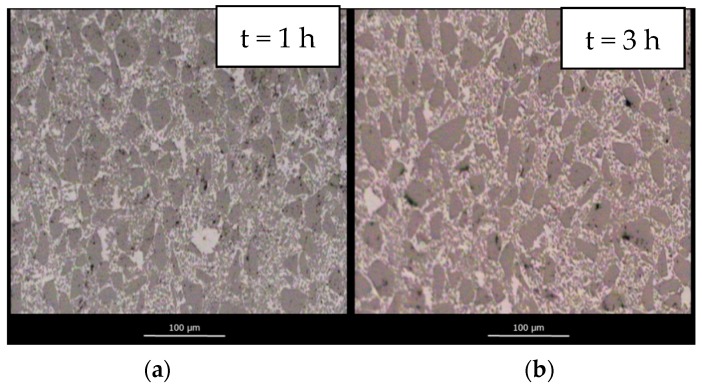
Evolution of the microstructure of SiC_p_/Si composite materials infiltrated at 1450 °C under a vacuum atmosphere with a 1 h (**a**), 3 h (**b**), 5 h (**c**) and 8 h (**d**) dwell time.

**Figure 10 materials-12-02425-f010:**
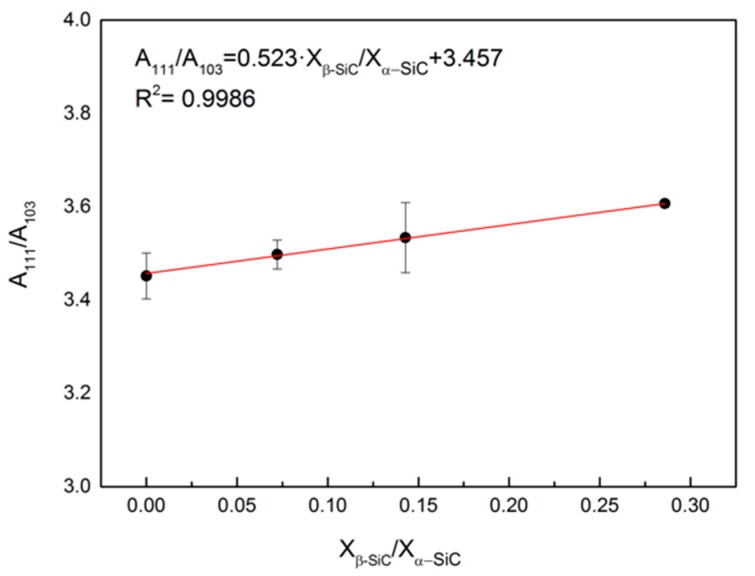
Calibration curve obtained from evaluating the relationship between the area below the 111 and the 103 diffraction peaks and the molar ratio between β-SiC and α-SiC.

**Figure 11 materials-12-02425-f011:**
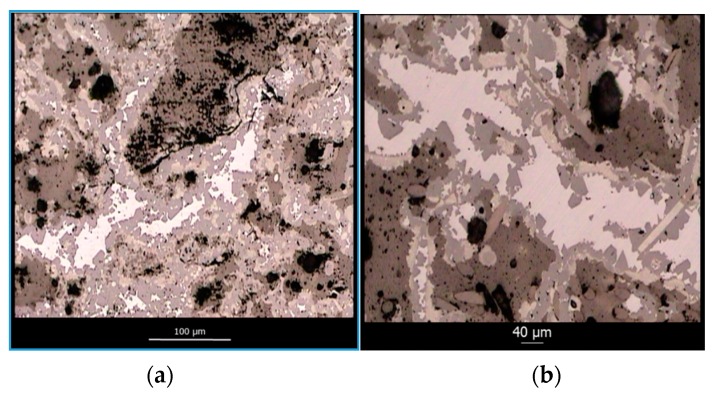
Representative micrographs of polished cross-sectioned C_f_/SiC specimens with a low (**a**) and a high (**b**) magnification.

**Figure 12 materials-12-02425-f012:**
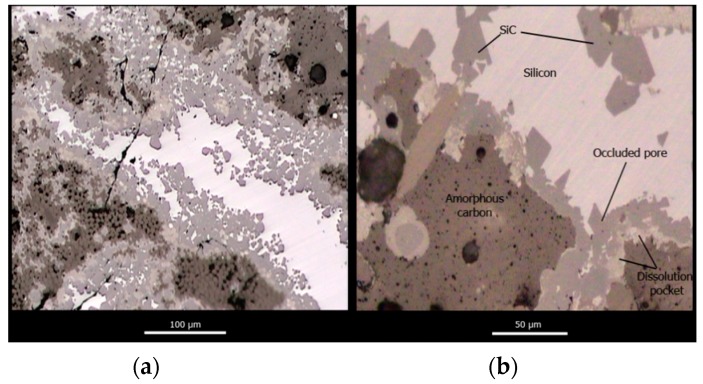
Pore reduction and pore closure phenomena in C_f_/SiC composite materials during reactive infiltration (**a**) and a zoomed in view of a totally occluded pore (**b**).

**Figure 13 materials-12-02425-f013:**
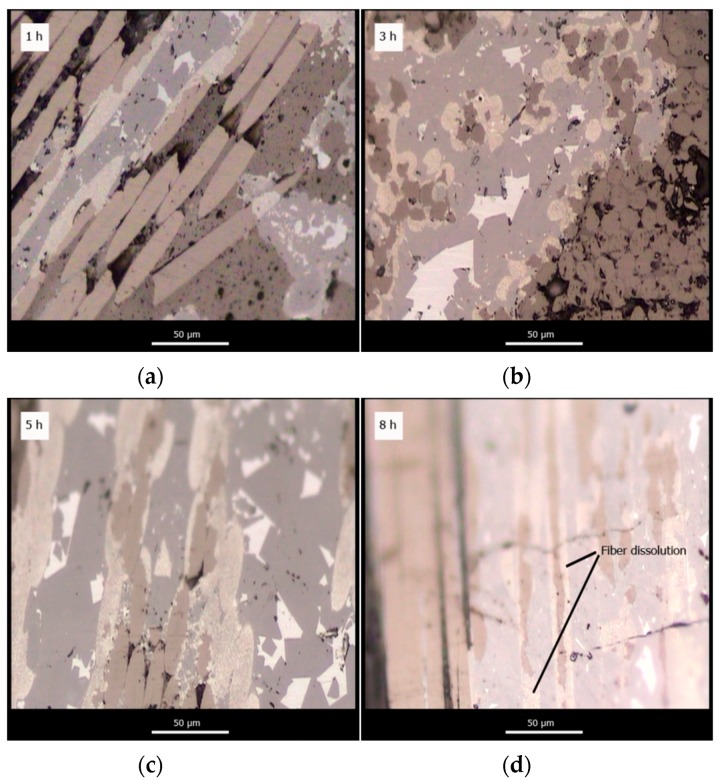
Evolution of carbon fibers during infiltration with a 1 h (**a**), 3 h(**b**), 5 h (**c**) and 8 h (**d**) dwell time.

**Figure 14 materials-12-02425-f014:**
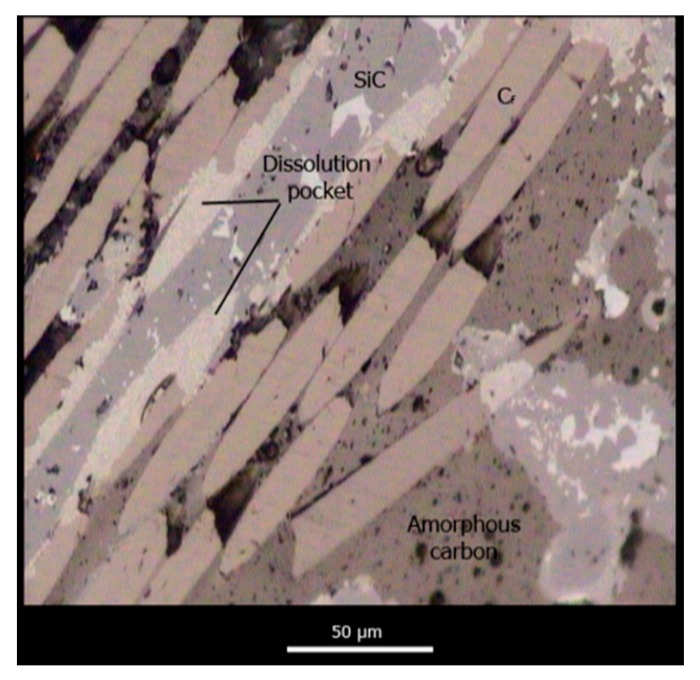
C_f_/SiC and C_f_/C interfaces in C_f_/SiC composite materials.

**Figure 15 materials-12-02425-f015:**
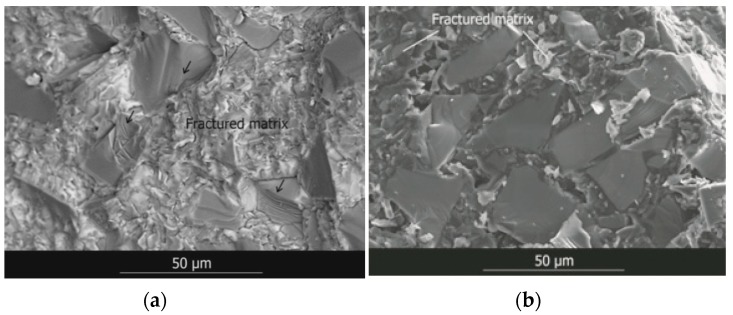
SEM image of the fracture surface of the SiC_p_/Si composite material after the 3-point bending test (**a**) and after the quasi-static compression test (**b**).

**Figure 16 materials-12-02425-f016:**
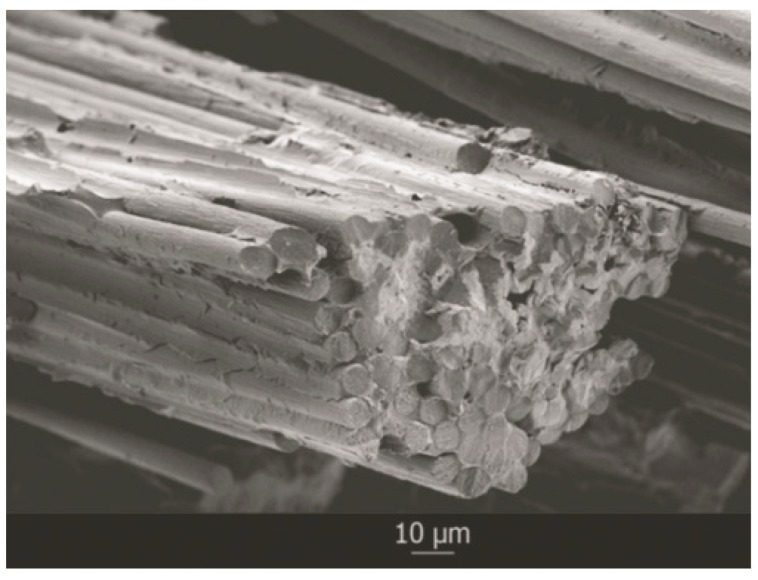
Fracture surface of the C_f_/SiC composite material after the 3-point bending test.

**Table 1 materials-12-02425-t001:** Results of the characterization of the SiC_p_/C preform.

Composition (wt.%)		Bulk Density (g/cm^3^)	He Density (g/cm^3^)	P (%)	D_50_ * (μm)
α-SiC	C	-			
95.30	4.70	2.10–2.20	3.10	30	1.30

***** D_50_: Average pore diameter measured by Hg porosimetry.

**Table 2 materials-12-02425-t002:** Results of the characterization of the C_f_/C preform.

Composition (wt.%)		Bulk Density (g/cm^3^)	He Density (g/cm^3^)	P (%)	D_50_ (μm)
C_f_	C	-			
70	30	1.20–1.25	1.83	30	13.40

**Table 3 materials-12-02425-t003:** Parameters used for the calculation of the infiltration time of liquid Si on the different preforms at 1450 °C.

Symbol	Magnitude	Value	SI Units	Reference
*λ*	Geometrical factor	2–4	-	[17,18,19]
a	Proportionality constant	4 × 10^−4^	-	[8]
$\gamma_{LV}$	Surface tension	0.750	N/m	[20,21,22]
*θ*	Contact angle of Si on C	40	°	[23]
η	Dynamic viscosity of liquid Si	0.605 × 10^−3^	Pa·s	[20]

**Table 4 materials-12-02425-t004:** Phase quantification of infiltrated C_f_/SiC composite materials.

Atmosphere	Dwell Time at 1450 °C (h)	% SiC	% Si	% C_f_ + C
Ar	1	37.6	5.3	57.1
	3	38.9	2.5	58.6
	5	38.8	7.0	54.1
	8	36.7	3.6	59.8
Vacuum	1	31.2	8.7	60.2
	3	30.0	7.7	62.7
	5	35.7	1.9	61.5
	8	35.0	3.5	62.7

**Table 5 materials-12-02425-t005:** Measured mechanical properties for the SiC_p_/Si composite material.

Material	Bending Strength (MPa)	Compressive Resistance (MPa)	Elastic Modulus (GPa)	Density (g/cm^3^)
SiC_p_/Si	310	520	110	2.8

**Table 6 materials-12-02425-t006:** Mechanical properties measured for the C_f_/SiC composite material infiltrated under optimum conditions.

Sample	Bending Strength (MPa)	Vickers Microhardness (HVN)	Elastic Modulus (GPa)	Density (g/cm^3^)	Friction Coefficient
C_f_/SiC	60	964	24	2.1	0.49

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
