# Peer review of "Key Parameters in the Manufacture of SiC-Based Composite Materials by Reactive Melt Infiltration"

_materials, 2019, doi:10.3390/ma12152425_

Round 1
Reviewer 1 Report
The paper by Narciso et al. deals with the synthesis of SiC composites and their properties. Overall the paper is clear and well presented. I recommend publication after minor revisions:
- in the materials and methods section authors re-direct to previous papers for details. This is ok, but I think that a brief description should be added also in the paper in order to facilitate the reader
- minor typos are present all over the manuscript, i.e. but not only on lines 13, 43, 138 and 429
Author Response
Dear reviewer, first of all thank you for the time spent and your valuable comments.
All the suggested changes have been made.

Reviewer 2 Report
This paper reports the influence of different process' parameters (dwell time, atmosphere) on the formation of silicon carbide composites by an infiltration techinque, highlithing the microstructure and mechanical properties, by means of several characterizations. The study is rather exhaustive and the conclusions are well supported. I suggest only some minor checks, in particular in the english text in the firt part (abstract, introduction) and to verify these aspects:
at page 7 there is probably something wrong with the references to the figures: it is cited figure8, that appears later in the text. (from row 206 to row 208 the references have to be verified)
at page 11 the labels on figure 9 are out of the image.
i also suggest to specify the meanings of SiCp and Cf, it would be helpful to a wide range of readers
Author Response

(The authors gave the same response as above.)

Reviewer 3 Report
The authors investigated the influence of dwell time and atmosphere during liquid silicon infiltration of two different precursors. The presentation is clear, nevertheless, I suggest minor revision of the manuscript. The following points should be attributed:
line 8: Affiliation 4 is not linked to an author
line 19, 20 etc.: Please explain abbreviations at first use
line 95: Please add a reference
line 112: Which resin was used
line 135: (10-3 s-1) --> is this the strain rate? --> please explain in the text
line 137: in which strain range did you calculate the modulus
line 321: Figure 11, not figure 16
line 379-382: The optimized parameters are the same for both samples. From your writing this is not clear immediately and readers have to check it again. Please change to a clear statement like you already did in the abstract and conclusions
line 412: the sample name should be Cf/SiC not SiCp/Si
line 418-419 + 434-438: Please add a more detailed discussion why these materials are suitable for the mentioned applications including values for commercial reference materials and/or data from other studies.
Author Response

(The authors gave the same response as above.)
